# Exploration of the Acceptance of the Use of Procalcitonin Point-of-Care Testing and Lung Ultrasonography by General Practitioners to Decide on Antibiotic Prescriptions for Lower Respiratory Infections: A Qualitative Study

Daniel Geis,[1] Nina Canova ![ORCID],[1] Loïc Lhopitallier,[2,3] Andreas Kronenberg,[4,5] Jean-Yves Meuwly,[6] Nicolas Senn,[7] Yolanda Mueller,[7] Fabienne Fasseur,[1] Noémie Boillat-Blanco[2]

DG and NC contributed equally. FF and NB-B contributed equally.

For numbered affiliations see end of article.

**Correspondence to**
Nina Canova;
canovanina7@gmail.com

## ABSTRACT

**Objectives** We aimed to explore the acceptance and opinions of general practitioners (GPs) on the use of procalcitonin point-of-care and lung ultrasonography for managing patients with lower respiratory tract infections in primary care. We suppose that there are several factors that can influence the physician's antibiotic prescribing decision, and the implementation of a new tool will only be possible when it can be inserted into the physician's daily practice, helping him/her in the decision-making process.

**Design** Semistructured interviews; data analysis using the grounded theory method.

**Setting** Lausanne, Switzerland.

**Participants** 12 GPs who participated in the randomised clinical trial UltraPro, which evaluated the impact of the use of procalcitonin only or an algorithm combining procalcitonin and lung ultrasonography on antibiotic prescription.

**Results** GPs had mostly positive attitudes towards the use of point-of-care procalcitonin in lower respiratory tract infections and uncertainties regarding the usefulness of ultrasonography. Physicians' prescribing decisions result from interactions between three kinds of TrustS (core category): 'self-confidence', 'trust in the results' and 'trust in the doctor–patient relationship'. Procalcitonin reinforced the three levels of trust, while ultrasonography only strengthened the physician–patient relationship. To facilitate implementation of procalcitonin, physicians pointed out the need of coverage by insurance and of clear guidelines describing the targeted patient population.

**Conclusions** Our data show that there is a preference for the implementation of procalcitonin rather than lung ultrasonography for the management of patients with lower respiratory tract infections in primary care. Coverage by insurance plans and updated guidelines are prerequisite to the successful implementation of procalcitonin testing in primary care.

**Trial Registration number** NCT03191071

## STRENGTHS AND LIMITATIONS OF THIS STUDY

⇒ The use of semistructured qualitative interviews elicited rich and complex data that offer important insights into the opinions of general practitioners (GPs) on new tools in medical practice.

⇒ Inclusion of GPs who used procalcitonin and lung ultrasonography reinforces our findings.

⇒ The study included GPs participating in the UltraPro trial. Therefore, this qualitative evaluation might not be representative of the whole Swiss GP population.

⇒ The GPs' willingness to participle in such kind of study reflects awareness of antibiotic resistance and interests in improving and questioning their practice.

## INTRODUCTION

Antimicrobial resistance, driven by decades of use, represents a threat to public health.[1] Several quantitative studies showed a positive impact of diagnostic test-based interventions on antibiotic prescription.[2–4] Although a new test might have a positive effect in the setting of controlled trial, large-scale implementation is usually followed by diminishing adherence and impact.[5] To ensure an enduring impact outside the scope of clinical research, it is important to investigate the acceptance and feasibility of the implementation of these tools by the professionals expected to use them.[6–9]

A recent randomised controlled study (the UltraPro Study) showed that point-of-care procalcitonin decreases antibiotic prescription in patients attending their general practitioner (GP) for a lower respiratory tract infection (LRTI) when compared with usual

**Table 1** Description of the three study groups, the test used and the training content in each group

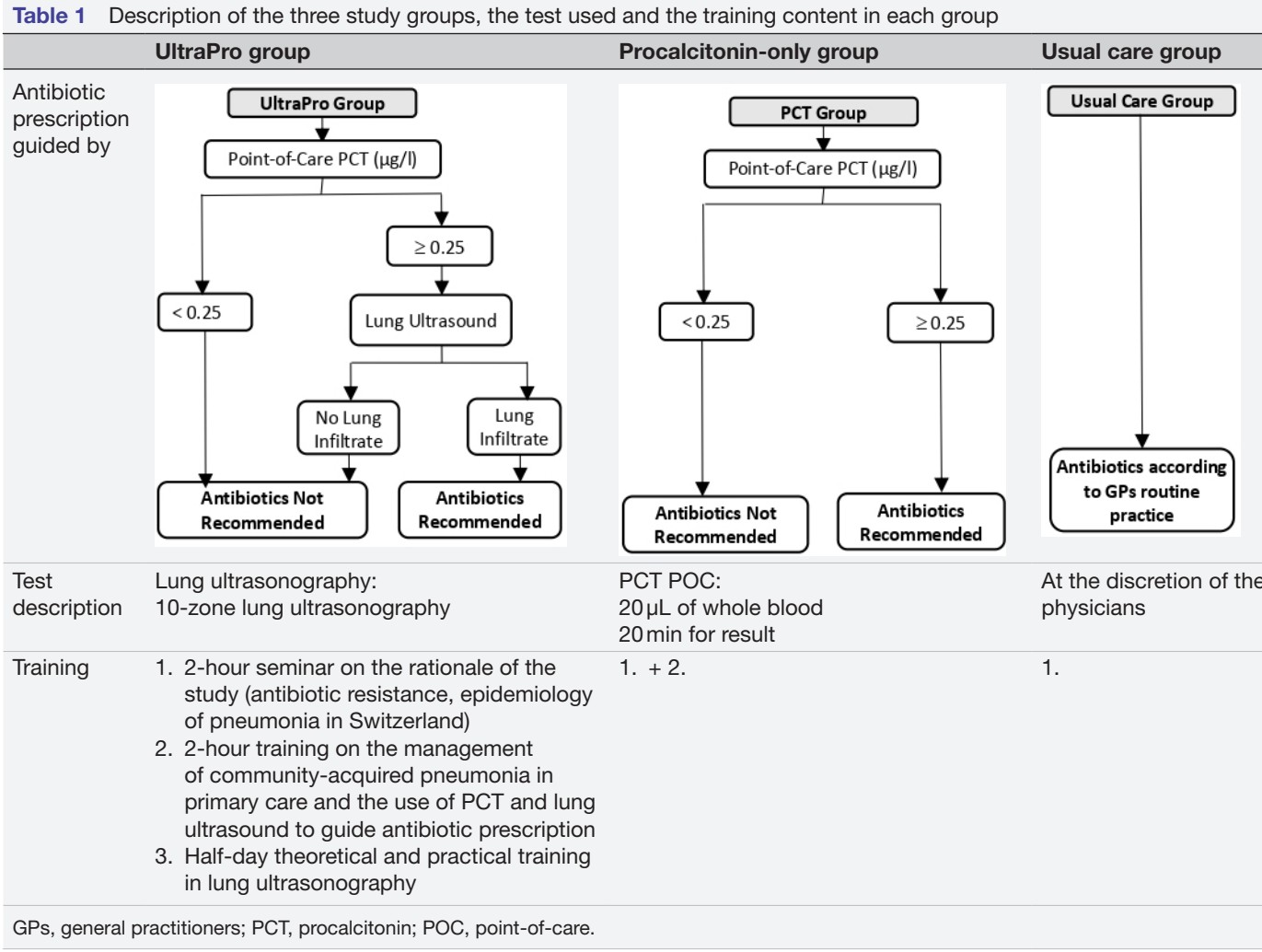

| | UltraPro group | Procalcitonin-only group | Usual care group |
|---|---|---|---|
| Test description | Lung ultrasonography: 10-zone lung ultrasonography | PCT POC: 20 µL of whole blood 20 min for result | At the discretion of the physicians |
| Training | 1. 2-hour seminar on the rationale of the study (antibiotic resistance, epidemiology of pneumonia in Switzerland) 2. 2-hour training on the management of community-acquired pneumonia in primary care and the use of PCT and lung ultrasound to guide antibiotic prescription 3. Half-day theoretical and practical training in lung ultrasonography | 1. + 2. | 1. |

GPs, general practitioners; PCT, procalcitonin; POC, point-of-care.

care.[10] A sequential algorithm (UltraPro) combining procalcitonin first and lung ultrasonography did not further decrease prescription. This study offers a unique opportunity to get more insights into the experience of GPs who used procalcitonin and lung ultrasonography during the intervention period. Indeed, few studies evaluated the perceptions of physicians using host biomarkers and ultrasonography and the result could support a tailored implementation of these new tools.[6 7 11–13] In order to identify barriers and facilitators to the implementation of these tests, we aimed to describe GPs' opinions of using procalcitonin and lung ultrasonography during the UltraPro clinical trial.

## METHODS
### The UltraPro trial
The UltraPro Study is a three-group cluster randomised controlled trial conducted between 6 September 2018 and 10 March 2020 in 60 GP practices in Switzerland as previously described.[14] Table 1 shows the intervention and the training content of the three study groups.

### The population sample of the qualitative study
The study population consists of a predefined sample of 12 French-speaking GPs selected among the 60 GPs who participated in the trial. The same number of GPs was selected from each study group of the UltraPro Study. We decided on this minimal sample size based on previous qualitative articles in the same setting showing data saturation by the 12th interview.[6 11] By the analysis of the 12th interview, no new items within categories emerged and we decided not to recruit additional GPs.

GPs were contacted by email and phone to plan the interviews.

Table 2 shows the characteristics of the 12 GPs who participated in this study. Their characteristics are comparable with the other GPs in the UltraPro Study.

### Study design: qualitative interviews and grounded theory method analysis
We conducted a qualitative study design based on the theoretical and methodological principles of the grounded theory method (GTM).[15 16]

First, two health psychology students (DG and NC) conducted face-to-face, individual semistructured

**Table 2** Characteristics of the general practitioners

| | General practitioners who participated in the qualitative study, N=12 | General practitioners who participated in the intervention study, N=48 | P value |
|---|---|---|---|
| Female, n (%) | 6 (50) | 19 (40) | 0.532 |
| Francophone, n (%) | 11 (92) | 39 (81) | 0.670 |
| >10 years' practice, n (%) | 6 (50) | 23 (48) | 1.000 |
| >5 general practitioners in practice, n (%) | 3 (25) | 9 (19) | 0.692 |
| Practice in urban setting, n (%) | 8 (67) | 33 (69) | 1.000 |
| Radiology available in practice, n (%) | 7 (58) | 28 (58) | 1.000 |
| Ultrasound available in practice before the study, n (%) | 3 (25) | 13 (27) | 1.000 |
| Procalcitonin use before the study, n (%) | 0 (0.0) | 1 (2.1) | 1.000 |

P value was calculated by Pearson's $X^2$ test or Fisher's exact test, as appropriate.

interviews, under the expertise of a lecturer (FF), in autumn–winter 2019–2020. An interview guide contained open questions to explore GPs' representations of the use of procalcitonin and lung ultrasonography and how they affect antibiotic prescription and the physician–patient relationship. We investigated the possible barriers, resources and proposals that could either help or hinder the implementation of procalcitonin and lung ultrasonography at the point-of-care in primary care. After the first interview, DG and NC analysed the adequacy of the interview grid, in relation to the relevance of the contents of the discourse. After an interjudge agreement with FF, all interviews were done with the same grid. The interviews lasted about 45 min (range of 34–56 min).

Second, DG and NC transcribed the recorded interviews. Third, DG and NC carried out an analysis of the data following the three stages of coding of GTM: open, axial and selective coding.[16] Simultaneously, after the interviews and the transcripts, they wrote memos to explore their reflexivity, feelings and links to their knowledge on the data.[15] They read and coded the transcripts independently and made another interjudge agreement after the elaboration of the categories including codes with similar meaning. This process enabled them to build the emerging theoretical framework from the data and to elaborate the core category. The systematic writing of the memos and the interjudge agreement allowed them to the observation of 'data saturation', which means that no new items were found by interviewees.

### Patient and public involvement
No patient involved.

### RESULTS
Globally, GPs (11 of 12) are aware of antibiotic resistance and its consequences and were keen to optimise their prescribing practice by using new tools.

The GTM coding resulted in three main categories (table 3):

1. Opinions on the use of procalcitonin point-of-care test (2 codes and 5 subcodes).
2. Opinions on the use of lung ultrasonography (2 codes and 5 subcodes).
3. Prescription decision factors (2 codes and 11 subcodes).

### Category 1: opinions on the use of procalcitonin point-of-care test
In all groups, the majority of GPs (10 of 12) showed a preference for the implementation of procalcitonin over lung ultrasonography for practical and cost reasons in primary care.

> I don't know… if the results show that it's more powerful to use ultrasound and procalcitonin than procalcitonin alone. If we have to use both tests to get the best result, I would do that. But if there is no need I would find it easier (laughs).
>
> *So you would prefer to use procalcitonin?*
>
> Yeah. On its own. Yeah, yeah. (MED8.115–116)

The majority of the GPs (seven of eight) of the procalcitonin and of the UltraPro groups believe that procalcitonin provides an objective result that can facilitate their practice. They mention that it is an easy-to-use and trustworthy tool which allows them to distinguish between a viral and a bacterial infection in a more reliable way than using other tests currently available. A typical comment included:

> Well, I think it's positive to say that we have a test which gives us a clear answer to a clear question […], I see that as positive […] (MED5.86)

All GPs (12) also perceive that this test has the potential to reassure patients about a management decision thanks to its objective and factual character. A typical quote included:

> […] That's the usefulness of having tools … even if clinical judgment works when people trust it. To have a tool like that, which allows us to give a

**Table 3** Subcodes, codes, categories and core category

| Subcode | Code | Category | Core category |
|---|---|---|---|
| Diagnosis support | Positive opinions on the use of procalcitonin point-of-care test | Opinions on the use of procalcitonin point-of-care test | **TrustS**<br>► Self-confidence<br>► Trust in the results<br>► Trust in the physician–patient relationship |
| Objective result | | | |
| Ease of use | | | |
| Time constraints | Negative opinions on the use of procalcitonin point-of-care test | | |
| Cost constraints | | | |
| Safety of use | Positive opinions on the use of lung ultrasonography | Opinions on the use of lung ultrasonography | |
| Low cost | | | |
| Patient reassurance | | | |
| Time constraints | Negative opinions on the use of lung ultrasonography | | |
| Subjective result | | | |
| Medical appreciation | Physician–patient relationship | Prescribing decision factors | |
| Patient's medical history | | | |
| Patient's education | | | |
| Patient's beliefs | | | |
| Patient's trust | | | |
| Professional experience | Physicians' prescription skills | | |
| Impact of training | | | |
| Clinical examination | | | |
| Follow-up consultation | | | |
| Delayed prescription | | | |
| Time constraints | | | |

quantitative result … it's very reassuring for everyone […] (MED6.70)

According to GPs (10 of 12), the procalcitonin test seems to have the potential to change the prescribing decision of the GPs and improve their practice.

Nine physicians pointed out some barriers to the use of procalcitonin. First, the health insurance does not cover its cost and, second, the time delay needed to obtain the result.

…the barrier I see is that it takes a while to get a result. (MED5.52)

[…] I think one question is how much it costs and then, the other question is whether it can be billed when measured in the practice, because if I can't bill it and I have to send it to the external lab it's a worthless test. This test is only worth if I can do it point-of-care and I can bill it point-of-care. (MED7.54)

Most GPs think (10 of 12) that this tool should not be used systematically in all consultations for LRTI. They mention that this test should mainly be used in select patient populations, for whom they have more doubts and thus tend to overprescribe antibiotics, such as the elderly, immunosuppressed or patients with medical comorbidities. Another group are patients who need additional arguments to be reassured about the doctor's decision not to prescribe.

… it depends a bit on the consultation we have. I have… I have quite a few elderly people for whom I go more easily for the antibiotic… even if it's not always *lege artis*. And that's where procalcitonin could help us. (MED9.8)

### Category 2: opinions on the use of lung ultrasonography
The majority of the GPs (9 of 12) say that the main advantage of this tool is that it is safe and non-invasive, as well as the image format result, which can reassure the patient.

Yes yes … if we can do a non-irradiant examination even if chest Xray does not irradiate the lungs much … but if we have an examination that is completely safe I think that yes … and moreover … more efficient. (MED1.69)

A minority of GPs were in favour of its use (5 of 12). These were mainly GPs who already used ultrasound for other purposes in their practice.

No, I use it for sports medicine… for the locomotor system. I find it fabulous because you can also explain it and show images to people. And as an argument for someone who is perhaps … who has a little preconceived idea about antibiotics … if I can show something on the echo, it's very powerful information. (MED3.141)

All GPs say that the main barrier to the use of ultrasound is its subjective and operator-dependent nature and the fact that it requires significant training and practice to obtain interpretable images. Most GPs (8 of 12) feel that even with extensive training, it would be difficult to rely on this test completely. A typical quote was:

Ultrasound is still linked to the person who does it, so they are people who will question the reliability of the test because the person is not well trained, or does not know how to interpret the images well […] (MED12.93)

In addition, most GPs (9 of 12) do not trust their own interpretation of ultrasonography images.

[…]the problem with ultrasound is that for me there is a problem of confidence in my own … my own skills in doing ultrasound. (MED11.117)

In private practice, seven GPs mention that it is complicated to use ultrasound systematically because of the limited time between consultations.

Yeah it takes a crazy amount of time. So it's true that … with patients every 15 minutes … using the ultrasound to look for… where there is a bit of oedema which then corresponds to a pulmonary condensation […] (MED8.104)

Three GPs suggest that well-trained doctors could do the ultrasonography examination.

So I am not sure it's right to implement this tool in all practices. If a doctor, who is experienced in ultrasound, is referent for ultrasound in a group practice, why not. […] (MED10.97)

### Opinions on the use of point-of-care tests

GPs see a benefit in the possibility of having a set of point-of-care diagnostic tools to help them in their decision-making. These tools could allow them to change their mind in certain situations, to reassure themselves in case of diagnostic doubt, as well as to reassure their patients. A typical quote included:

[…] there are some patients who are positively influenced by the use of technology, let's say. Patients who will feel better cared for if such tools are used. (MED8.112)

To facilitate a successful implementation of new diagnostic tests, five GPs think it necessary to establish clear guidelines describing the targeted patient population and the recommended clinical follow-up to evaluate the evolution of the disease over time and avoid complications.

[…] Maybe propose to see the patient again and do the examination in … in a few days or … […] no antibiotic but maybe sometimes, there are situations that are worth being … monitored a little bit … knowing that finally an algorithm will never be 100% …

maybe just to have a particular management of those patients who may still benefit from the antibiotic … under certain conditions. (MED1.72)

### Category 3: prescription decision factors

Most physicians (11 of 12) say that they are vigilant about their antibiotic prescribing levels. Basic and postgraduate training, being part of quality circles and studies, are important aspects that play a role in their sensitivity to the topic of antibiotic resistance. A typical quote was:

Oh yes ((laugh)), we're being asked to try to be careful with antibiotics. We've done training. (MED11.10–12)

Five GPs mention that their clinical findings are more important than test results and may lead them to disregard the result. Tests are only a part of the prescription process but not the biggest one.

[…] so we know that algorithms are not 100% anyway. But that's part of the practice of everything we do. So we do … we trust the algorithm as much as we trust any medical test knowing that nothing is perfect. (MED1.43)

Complementary examinations are therefore put in second place. These tests can be useful when there are doubts about the differential diagnosis, particularly with fragile patients. A typical quote included:

[…] sometimes there are situations that are worth being … to be monitored a little bit … knowing that finally an algorithm will never be 100% … just to have a close management of these patients. (MED1.72)

For all physicians, patients' beliefs are strong and may impair the prescription process through the patient–physician relationship. Patients may have misconceptions of treatment and, sometimes, unrealistic knowledge about antibiotics, which are difficult to overcome.

[…] There are people who have been used to receive antibiotics and have always felt that it worked very well … for them it is difficult.

*Mhm.*

So these people are very difficult to convince that there's no need to take antibiotics. And it's true that sometimes we prescribe antibiotics … ((laughs)) if the patient insists a lot. (MED4.19–20)

For eight physicians, the level of trust in the relationship influences the patient's responsiveness to explanations and discussion with the physician regarding the prescription of antibiotics.

Well, they are … I think they trust us. If we tell them "you have to take it" they take it. And then if we tell them "it's not necessary" they are often convinced … that it's not necessary. Most patients follow our advices quite well. (MED4.28)

All physicians mention their education role towards their patients. Indeed, they think that it is necessary to take time to educate patients to make them understand, for example, that antibiotics do not cure all diseases.

> Sometimes we talk about viruses and bacteria … some of them don't know the difference. We say that not everything is cured with an antibiotic … and that antibiotics also have side effects … sometimes they are not aware that antibiotics can have severe side effects […] (MED1.80)

Nine physicians see patients as partners. They mention that the mentality of patients is changing and they are becoming more aware of the problem of antibiotic resistance and listening to the advice of physicians.

> […] And then these patients well … I think that there is a little bit of … media hype. They have read some campaign … and … emh … public opinion is changing. Compared with when I started my internship in 2005 … I think it has already changed a lot. Emh … and the patients are ready to hear our arguments in general. (MED8.29)

Four physicians say that delayed prescription constitutes an usual strategy to empower patients and to avoid overprescription.

> […] either we tell them, well, if we really feel they are very angry, for example, it happens, some of them are angry, ehm … I tell them, well, look, I'll give you the prescription … you think about it … but that the

two or three times that were hesitant didn't take the antibiotics anyway - and they were happy because they got better without. (MED5.41)

From our GTM analysis, antibiotic prescribing results from a combination of three kinds of trusts for physicians. TrustS is our core category (figure 1) englobing (1) trust in the physician's own subjectivity, which we call 'self-confidence'; (2) trust in the objective test data, which we call 'trust in the results'; and (3) trust in the relationship between the consultant and the consulted, which we call 'trust in the physician–patient relationship'.

## DISCUSSION

Acceptance of a new diagnostic test depends on the influence of the tool on GPs' self-confidence, GPs' trust in the results and physician–patient relationship. We observed that procalcitonin reinforces the three levels of trust, while ultrasonography only strengthens the physician–patient relationship. We identified mostly positive attitudes towards the use of point-of-care procalcitonin in LRTI contrasting with uncertainties about the usefulness of ultrasound.

### Procalcitonin

The majority of GPs emphasised that the availability of a simple tool like point-of-care procalcitonin facilitates their practice and has the potential to change their antibiotic prescription decision by increasing their confidence. Another study evaluated hospital physicians' experiences

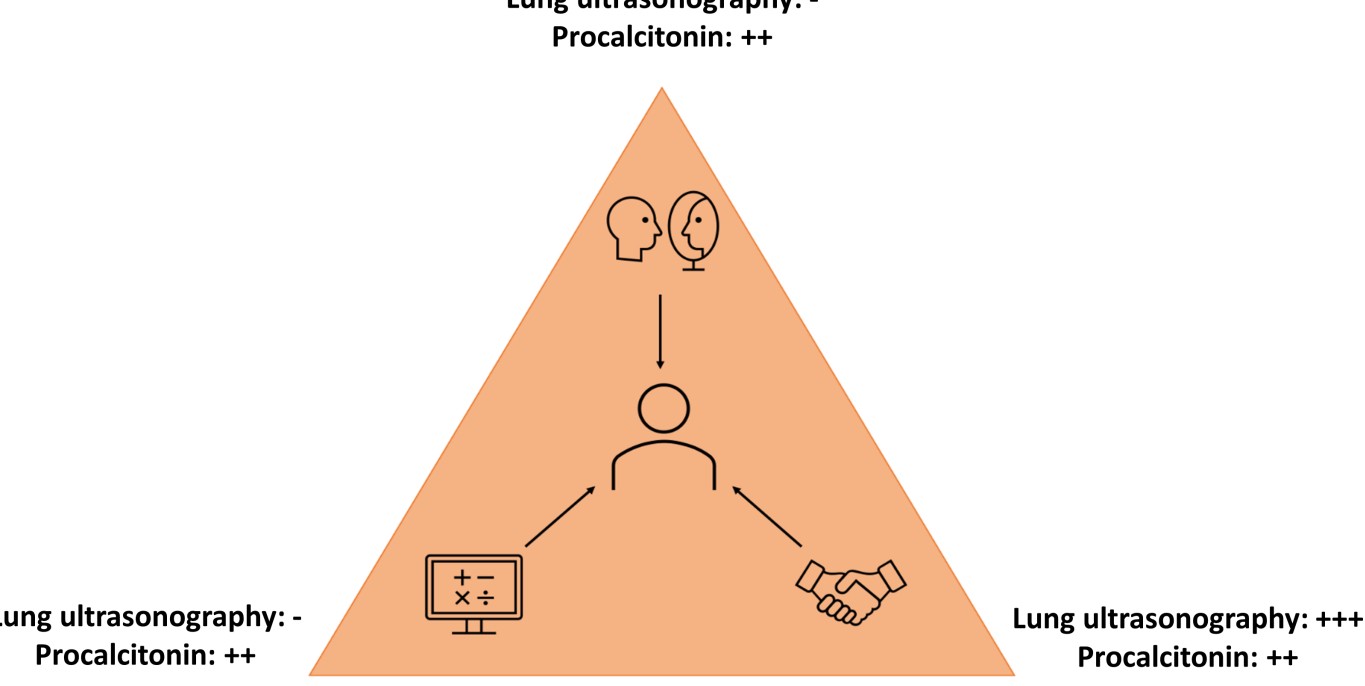

**Lung ultrasonography: -**
**Procalcitonin: ++**

**Lung ultrasonography: -**
**Procalcitonin: ++**

**Lung ultrasonography: +++**
**Procalcitonin: ++**

**Figure 1** Representation of the concept of trusts behind antibiotic prescription and the impact of procalcitonin and lung ultrasonography. A complex interaction between three types of trust influences the physician (in the centre) in his decision to prescribe antibiotics: self-confidence (top image), trust in the results (left image) and trust in the physician–patient relationship (right image). **-**, absence of impact on the trust; **+**, reinforcement of the trust, strong trust.

with procalcitonin and showed that a knowledge gap made physicians uncertain about the appropriate use of procalcitonin, its interpretation and trustworthiness.[11] These results show the importance of training when implementing a new diagnostic tool to improve adherence to the guidance.

However, our results are in line with a previous qualitative analysis evaluating the experience of GPs with the use of point-of-care C reactive protein (CRP) for LRTIs. This study showed that CRP empowered GPs to prescribe less antibiotics.[6] Our findings are not specific to procalcitonin, but rather to the use of a biomarker at the point-of-care to guide clinicians.

Another finding is the use of the procalcitonin result as a support in the discussion with patients who wish for antibiotics, to reassure them about the prescription decision. This is in line with the results of a study directly exploring patients' views of primary care consultations using point-of-care CRP and/or training communication skills.[17] Patients perceived that both interventions provided GPs with support regarding their decision to prescribe or not antibiotics and improved their understanding of this decision. A study in the UK also showed the potential value of a point-of-care test to convince patients about the antibiotic prescription decision.[7] Other qualitative studies describe the impact of a test on patients' reassurance about the management decision.[18 19]

We also identified barriers to the use of procalcitonin, mainly concerns regarding its cost and coverage by health insurance. GPs mentioned that they would not use the test if they cannot bill it as a point-of-care test or if they have to send it to a laboratory with a delayed result. The same argument was identified in a qualitative study evaluating CRP point-of-care in a similar population.[6] GPs also asked for guidance to select which patients should be tested to avoid testing patients without a clear expected benefit. A previous study also highlighted the importance of clearly defining the indication before testing and of updating guidelines before implementation.[6]

### Lung ultrasonography

GPs appreciate the innocuous nature of ultrasonography and the positive impact it has on patients. A qualitative study exploring GPs' experiences of using ultrasonography in primary care also showed that the visual nature of examinations provides reassurance to patients.[12] Another qualitative study conducted in the emergency department and investigating patient confidence in the physician showed that the use of ultrasonography improved the quality of the physician–patient relationship.[20]

However, the majority of GPs were sceptical about the implementation of ultrasonography. The main identified barriers were the subjectivity of its interpretation and its operator-dependent nature, which lead users to feeling uncomfortable. This finding contradicts a previous study exploring perceptions of ultrasound use by GPs.[12] In this study, all interviewed GPs used ultrasound frequently (monthly to daily use) to answer simple clinical questions

and felt comfortable with its interpretation, adding that ultrasound examinations provided them with a sense of reassurance. However, they also performed more explorative ultrasound examinations outside their catalogue and were insecure about their findings. These data underline the importance of integrating ultrasonography in the consultation as a continuum and supplement to the patient's history and physical examination. Using ultrasonography rarely, as it was the case in our study, does not allow the physician to reach a comfort zone even for focused examination addressing simple clinical questions. Indeed, in the UltraPro Study, ultrasonography was only recommended in patients with an elevated procalcitonin, which happened in only a minority of patients. Another barrier is related to the training in the use and interpretation of this tool. A recent review on the use of point-of-care ultrasonography in general practice highlighted the variety in the length of the training programmes (between 2 and 320 hours) and concluded that we need further assessment of the quality of ultrasonography to identify the optimal training of GPs.[12] A systemic review showed that, in the majority of studies, lung ultrasonography in the hands of non-imaging specialists to diagnose pneumonia had a high sensitivity and specificity.[21] However, the heterogeneity between studies prevents from defining the optimal training format. In our study, the half-day training did not allow GPs to have enough trust in their ability to use this tool, although it may also be due to the infrequent use of the tool in their practice.

The use of semistructured qualitative interviews and GTM analysis elicited rich and complex data that offer insight into the acceptance and opinions of GPs on new tools in medical practice. However, our research has some limitations. The first one is the limited number of participants. Second, the quality and spontaneity of our interviews in the GP practices, which were undermined by interruptions like phone calls or questions from the medical assistants. Third, GPs participating in the UltraPro trial, as well as in this qualitative evaluation, might not be representative of the whole Swiss GP population. Their willingness to participle in such kind of study reflects awareness of antibiotic resistance and interests in improving and questioning their practice.

### CONCLUSION

This qualitative study adds to and complements the results of the UltraPro clinical trial. It identified three levels of trust that affected acceptance of a new diagnostic test targeting antibiotic prescription. While procalcitonin led to an empowerment of the GPs through a positive impact on the three levels of trust, ultrasonography was difficult to integrate into these levels of trust and its subjective interpretation was seen as a negative point. Our data show that there is a preference for the implementation of procalcitonin rather than lung ultrasonography for the management of patients with LRTIs in primary care. This echoes the quantitative results of the UltraPro trial,

and this convergence of our qualitative and quantitative analyses highlights the need for the development of clear guidance on procalcitonin point-of-care use, together with adequate training and reimbursement of all required for its successful implementation.

By contrasting a qualitative approach, which focuses on the subjectivity and singularity of the experience, to the quantitative methodology, which is more objective and aims to identify general operating rules, we were able to better appreciate how innovations in primary care could be implemented, and thus have a real impact on patients' outcomes and public health.

**Author affiliations**
[1]Research Center for Psychology of Health, Aging and Sport Examination (PHASE), University of Lausanne, Lausanne, Switzerland
[2]Infectious Diseases Service, Lausanne University Hospital, Lausanne, Switzerland
[3]gare10 medical practice, Lausanne, Switzerland
[4]Institute for Infectious Diseases, University of Bern, Bern, Switzerland
[5]Medix General Practice, Bern, Switzerland
[6]Department of Radiology, Lausanne University Hospital, Lausanne, Switzerland
[7]Department of Family Medicine, Center for Primary Care and Public Health, Lausanne, Switzerland

**Acknowledgements** We thank all general practitioners who accepted to participate and make this study possible. We are grateful to José Knusli who did table 2.

**Contributors** NB-B was the guarantor. NB-B, LL, YM, AK, J-YM and NS conceptualised and designed the project. DG, NC and FF designed the research protocol. NC and DG collected data, analysed the transcripts and led the manuscript development. NB-B, FF and LL provided detailed feedback on early iterations of the manuscript. All authors have seen, reviewed and approved the final version.

**Funding** This work was supported in part by the Swiss National Science Foundation (grant number 407240_167133) and by an academic award of the Leenaards Foundation (to NB-B).

**Disclaimer** The funders had no role in considering the study design or in the collection, analysis, interpretation of data, writing of the report or decision to submit the article for publication.

**Competing interests** None declared.

**Patient and public involvement** Patients and/or the public were not involved in the design, or conduct, or reporting, or dissemination plans of this research.

**Patient consent for publication** Not required.

**Ethics approval** This study involves human participants and was approved by the Swiss ethics committees of cantons Vaud and Bern (2017-01246). All GPs provided written consent to participate in this study.

**Provenance and peer review** Not commissioned; externally peer reviewed.

**Data availability statement** No data are available.

**ORCID iD**
Nina Canova http://orcid.org/0000-0002-7407-4577

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
