## [Reviewer comments · BMJ Open]

ARTICLE DETAILS

TITLE (PROVISIONAL)	Qualitative exploration of the acceptance of the use of procalcitonin point-of-care testing and lung ultrasonography by general practitioners to decide on antibiotic prescriptions for lower respiratory infections
AUTHORS	Geis, Daniel; Canova, Nina; Lhopitallier, Loïc; Kronenberg, Andreas; Meuwly, Jean-Yves; Senn, Nicolas; Mueller, Yolanda; Fasseur, Fabienne; Boillat-Blanco, Noémie

VERSION 1 – REVIEW

REVIEWER	Angela R. Branche Univ Rochester, Medicine, Division of Infectious Diseases
REVIEW RETURNED	06-Jul-2022

GENERAL COMMENTS	General Summary: The authors of this interesting manuscript used a semi-structured interview design to assess perceptions and experiences with the use of an algorithm incorporating ultrasonography and PCT testing to guide antibiotic prescribing practices in GP offices in Switzerland. While the subject matter and approach is very interesting and potentially could be informative, there were major issues in the design, description of the methodology and creation of the results that require major thought and revision. Please see below for specific comments. Abstract: In the objectives section the authors describe an aim to explore mechanisms leading to antibiotic prescription. I'm not sure mechanism is the right word since there is no real description of under what circumstances antibiotics might be prescribed or the detailed factors that influence practice. Introduction: Line 12 I believe the authors mean adherence rather than adhesion. Methods: Training for Group #2 (PCT), it's not clear why they would have a 2 hour training of both pneumonia, PCT and lung US when they were not using lung US as part of the protocol. Is this an error? The sample size of only 12 GPs is very small. Did the individual semi-structured interviews of individual providers occur sequentially? More description is needed here. There is no description of how the interview data is further coded nor description of the categories leading to the Core category. Results: There is no description of how the three Core categories of types of trust were determined. Are these established core categories, or ones that were created by investigators? If the latter, how were these categories constructed? There are a lot of statements like, page 8, line 3, "the majority of GPs showed a preference for..." or line 19, page 8 and line 40, page 9 where words like majority, minority, most are used. This is very
--

	vague and if the authors would like to quantify responses, this again would require some sort of coding of the data and a quantity (number or %) so that the reader can understand how strongly these perceptions resonated in this small sample size. The addition of the statements from the GPs are interspersed in what reads more like interpretive conclusions rather results. It would have been better to structure the results section based on the specific question in the interview and how answers were coded. Discussion: The discussion is well written and gives good interpretation of the data but has content that is similar to the results highlighting that the results is more interpretative than expository. On page 13, lines 49-55, the discussion of US use in the GPs office is very unclear. The authors write that use of US was rarely done. Please explain in methods and discussion the frequency of use of these tools before and after implementation of the study protocol.
--	---

REVIEWER	Fei Peng Zhongda Hospital, School of Medicine, Southeast University, Department of critical care medicine
REVIEW RETURNED	12-Jul-2022

GENERAL COMMENTS	 1. The study population is not representative and the promotion value is low 2. The sample size is too small and statistics are not available, resulting in low reliability of conclusions. 3. Scores or grades can be designed for responses to facilitate statistical analysis.
---

REVIEWER	Eileen Huang Vanderbilt University Medical Center
REVIEW RETURNED	14-Aug-2022

GENERAL COMMENTS	This study used qualitative methods to assess general practitioners' (GPs) perspective of using procalcitonin point-of-care testing and lung ultrasonography to make decision on antibiotic prescription for lower respiratory infections. The GPs were recruited from the UltraPro study. Results showed that insurance coverage and training played crucial roles in GPs' decision making. This study adds value to clinical decision making from a qualitative perspective, however, a few things can be improved upon:  1) Less than 50% of the GPs from the UltraPro study participated in this study. If possible, increase the sample size would make the conclusion more representative. In general, a sample size of at least 30 is needed to do a significant test. 2) In Table 2. Characteristics of the general practitioners, please indicate what statistical method was used to calculate the p-value. 3) Please be more precise when discussing GPs' perspective. For instance, instead of saying "A majority of the GPs...", please indicate the number of GPs said so and so. For example: "A majority of the GPs (9/12)...". Perhaps a table that lists all the key perspectives along with number of GPs would be helpful.
--

REVIEWER	Aline Sarradon-Eck Aix Marseille University, SESSTIM
REVIEW RETURNED	14-Dec-2022

GENERAL COMMENTS	As requested by the editor-in-chief, my review emphasises only the methodological aspects of the study.
---

	The qualitative methodology is clearly stated, but need to be more detailed. 1- Sampling: the authors argued the sample was formed by convenience and the GPs were “selected”. Please, specify the selection criteria. If the sample was formed by convenience, the characteristics of the study population are not a result, in contrast to a random sample. So please, insert the table 2 into the methodological section. You can add that you have been looking for GPs with characteristics comparable to the rest of the GPs (if this is what you did). 2- The sample seems too small to have a data saturation. But it is true that the analyses made of the interviews do not show very complex findings... Basing a sample size on data saturation implies the analysis began with the first interview in an iterative process. Please specify if you did this. 3- Data analysis needs to be more detailed. It is very hard from this methods section to know what you actually did to get to your findings. You state having used Grounded Theory approach. However, this is not obvious from the methodological section or results section. You seem go beyond coding and produce a theory of process with the section “mechanism leading to antibiotic prescription”, but: 1) please give a little more information about the steps you performed to increase the credibility of your analysis; 2) this part of your results is not enough detailed and did not integrate quotations. So it is not possible to verify your analysis. In contrast, the rest of the analysis seems to be a more superficial thematic analysis. More fundamentally, the results presented do not accurately reflect the objectives of the study:  - “to describe GPs’ experiences and perceptions” : the results presented are only GPs’ opinions on the tests. However, opinions may be sufficient to prove the usefulness of these tests. - “to investigate mechanisms leading to antibiotic prescription”: the Core Category (p.8, l. 46-55) is not enough detailed and discussed while the literature (health psychology and medical sociology) is extensive on the topic of antibiotic prescription and on the topic of trust. p.15, l. 19-21: “The strengths of our study are that the results suggest the value of health psychology in medical practice.” Please, be a little more modest! If your study has a value, its value is to shed light on the opinions of physicians. In other words its value here on research issues, not on medical practice
--	--

VERSION 1 – AUTHOR RESPONSE

Reviewer	Comments	Response
----------	----------	----------

#1 Dr. Angela R. Branche	General Summary: The authors of this interesting manuscript used a semi-structured interview design to assess perceptions and experiences with the use of an algorithm incorporating ultrasonography and PCT testing to guide antibiotic prescribing practices in GP offices in Switzerland. While the subject matter and approach is very interesting and potentially could be informative, there were major issues in the design, description of the methodology and creation of the results that require major thought and revision. Please see below for specific comments.	We thank the reviewer for the comment. We made major changes in the methods and results parts of the manuscript. We also added a table (Table 3) to facilitate the understanding of the results.
	Abstract: In the objectives section the authors describe an aim to explore mechanisms leading to antibiotic prescription. I'm not sure mechanism is the right word since there is no real description of under what circumstances antibiotics might be prescribed or the detailed factors that influence practice.	We agree with the reviewer that our main objectives were to explore the perceptions and acceptance of the use of procalcitonin point-of-care and lung ultrasonography by general practitioners. When exploring these points, we learned about some physicians' prescribing decision factors. We modified the objectives in the abstract, in the last paragraph of the introduction and in the methods part.
	Introduction: Line 12 I believe the authors mean adherence rather than adhesion.	Thank you for pointing out this mistake. We corrected the text accordingly.
	Methods: Training for Group #2 (PCT), it's not clear why they would have a 2 hour training of both pneumonia, PCT and lung US when they were not using lung US as part of the protocol. Is this an error?	Thank you for pointing out this mistake. We corrected Table 1 accordingly.
	The sample size of only 12 GPs is very small. Did the individual semi-structured interviews of individual providers occur sequentially? More description is needed here.	Based on the literature, we chose to conduct semi-structured interviews of 12 GPs. Indeed, two studies conducted in the same setting showed data saturation by the 12th interview (Cals et al, Family Practice 2010 & Christensens et al, BMC Infect Dis 2020). As we did not identify new items within our categories when analyzing the 12th interview, we did not recruit additional GPs. We provided more details in the methods part of the manuscript.
	There is no description of how the interview data is further coded nor description of the categories leading to the Core category.	We provided more information on the coding of interview data in the methods. In the results section we added a table (Table 3) describing the sub-codes, codes, category and core category to facilitate the understanding of the path between the codes and the core category.
	Results: There is no description of how the three Core categories of types of trust were determined. Are these established core categories, or ones that were created by investigators? If the latter, how were these	We created the core categories based on sub-codes, codes and categories identified in the transcripts. As stated above, table 3 provides more details on the creation of the core categories.

categories constructed?	
There are a lot of statements like, page 8, line 3, “ the majority of GPs showed a preference for...” or line 19, page 8 and line 40, page 9 where words like majority, minority, most are used. This is very vague and if the authors would like to quantify responses, this again would require some sort of coding of the data and a quantity (number or %) so that the reader can understand how strongly these perceptions resonated in this small sample size.	As requested, we quantified the responses to show the importance of the different results.
The addition of the statements from the GPs are interspersed in what reads more like interpretive conclusions rather results. It would have been better to structure the results section based on the specific question in the interview and how answers were coded.	We extensively reviewed the results part of the manuscript. It follows the categories, which are described in Table 3. We also removed some interpretive conclusions.
Discussion: The discussion is well written and gives good interpretation of the data but has content that is similar to the results highlighting that the results is more interpretative than expository.	As stated above, we extensively modified the results part of the manuscript.
On page 13, lines 49-55, the discussion of US use in the GPs office is very unclear. The authors write that use of US was rarely done. Please explain in methods and discussion the frequency of use of these tools before and after implementation of the study protocol.	We added an explanation for the rare use of ultrasonography in the UltraPro study: “Indeed, in the UltraPro study, ultrasonography was only recommended in patients with an elevated procalcitonin which happened in only a minority of patients.” This is also shown in Table 1, in the methods part of the manuscript.
1. The study population is not representative and the promotion value is low	This study is a qualitative study, which aims to complement the results of a clinical trial. As stated in the conclusion of the manuscript, this qualitative study focuses on the subjectivity and singularity of the experience, and we do not expect that it is representative of the whole swiss GPs population. We also mention this point in the limitations of the study. However, when comparing GPs who participated in this qualitative analysis to all GPs of the UltraPro clinical trial, we did not observe significant differences (Table 2).
2. The sample size is too small and statistics are not available, resulting in low reliability of conclusions.	Based on the literature, we chose to conduct semi-structured interviews of 12 GPs. Indeed, two studies conducted in the same setting showed data saturation by the 12th interview (Cals et al, Family Practice 2010 & Christensens et al, BMC

	Infect Dis 2020). As we did not identify new items within our categories when analyzing the 12th interview, we did not recruit additional GPs. We provided more details in the methods part of the manuscript. As it is a qualitative and not a quantitative study, there is no need to provide statistics.
3. Scores or grades can be designed for responses to facilitate statistical analysis.	As it is a qualitative study based on semi-structured interview, we do not perform statistical analyses.
This study used qualitative methods to assess general practitioners' (GPs) perspective of using procalcitonin point-of-care testing and lung ultrasonography to make decision on antibiotic prescription for lower respiratory infections. The GPs were recruited from the UltraPro study. Results showed that insurance coverage and training played crucial roles in GPs' decision making. This study adds value to clinical decision making from a qualitative perspective, however, a few things can be improved upon:	Thank you for the comment. Please, refer to response 2 of reviewer #2.
1) Less than 50% of the GPs from the UltraPro study participated in this study. If possible, increase the sample size would make the conclusion more representative. In general, a sample size of at least 30 is needed to do a significant test.	
2) In Table 2. Characteristics of the general practitioners, please indicate what statistical method was used to calculate the p-value.	P value was calculated by Pearson's chi-squared test or Fisher's exact test, as appropriate. We added this information in the footnote of Table 2.
3) Please be more precise when discussing GPs' perspective. For instance, instead of saying "A majority of the GPs...", please indicate the number of GPs said so and so. For example: "A majority of the GPs (9/12)...". Perhaps a table that lists all the key perspectives along with number of GPs would be helpful.	As requested, we quantified the responses to show the importance of the different results.
As requested by the editor-in-chief, my review emphasises only the methodological aspects of the study. The qualitative methodology is clearly stated, but need to be more detailed. 1- Sampling: the authors argued the sample was formed by convenience and the GPs were "selected". Please, specify the selection criteria. If the sample was formed by convenience, the characteristics of the study population are not a result, in contrast to a random sample.	We provided more details on the qualitative methodology in the methods part of the manuscript. Regarding the study sample, please refer to response 2 of reviewer #2. We moved table 2 in the methods part of the manuscript.

So please, insert the table 2 into the methodological section. You can add that you have been looking for GPs with characteristics comparable to the rest of the GPs (if this is what you did).	
2- The sample seems too small to have a data saturation. But it is true that the analyses made of the interviews do not show very complex findings... Basing a sample size on data saturation implies the analysis began with the first interview in an iterative process. Please specify if you did this.	We provided more details on the sample size and data saturation in the methods part of the manuscript. For sample size, please refer to response 2 of reviewer #2.
3- Data analysis needs to be more detailed. It is very hard from this methods section to know what you actually did to get to your findings. You state having used Grounded Theory approach. However, this is not obvious from the methodological section or results section.	We thank you for this comment, which has improved the quality of our manuscript. We provide more details on the data analysis process in the methods part of the manuscript. We provided more
You seem go beyond coding and produce a theory of process with the section “mechanism leading to antibiotic prescription”, but: 1) please give a little more information about the steps you performed to increase the credibility of your analysis; 2) this part of your results is not enough detailed and did not integrate quotations. So it is not possible to verify your analysis. In contrast, the rest of the analysis seems to be a more superficial thematic analysis.	information on the steps to reach our results. We added Table 3 to show how we built our core category. We slightly modified the aims of our qualitative study as our interview guide contained open questions to explore GPs’ opinions on the use of procalcitonin and lung ultrasonography and how they affect antibiotic prescription and the physician-patient relationship. But the aim was not to evaluate in details the mechanism leading to antibiotic prescription. We also adapted the results part of the manuscript accordingly.
More fundamentally, the results presented do not accurately reflect the objectives of the study: - “to describe GPs’ experiences and perceptions” : the results presented are only GPs’ opinions on the tests. However, opinions may be sufficient to prove the usefulness of these tests.	Based on your advice and to be more accurate with regards to our results, we modified the objectives of the study: to describe GP’s opinions on the tests.
- “to investigate mechanisms leading to antibiotic prescription”: the Core Category (p.8, l. 46-55) is not enough detailed and discussed while the literature (health psychology and medical sociology) is extensive on the topic of antibiotic prescription and on the topic of trust.	As stated above, our study did not aim to analyse the mechanisms leading to antibiotics prescription and we adapted our aims accordingly: to explore GPs’ opinions on the use of procalcitonin and lung ultrasonography and how they affect antibiotic prescription and the physician-patient relationship. We provided more details on data analysis and on the coding strategy as you can see in the methods and results parts and in Table 3

p.15, l. 19-21: "The strengths of our study are that the results suggest the value of health psychology in medical practice." Please, be a little more modest! If your study has a value, its value is to shed light on the opinions of physicians. In other words its value here on research issues, not on medical practice	We agree with the reviewer and modified the text to remove the sentence on the value of health psychology.
---	--

VERSION 2 – REVIEW

REVIEWER	Fei Peng Zhongda Hospital, School of Medicine, Southeast University, Department of critical care medicine
REVIEW RETURNED	01-Mar-2023

GENERAL COMMENTS	The sample size is too small and statistics are not enough, resulting in low reliability of conclusions.
--

REVIEWER	Eileen Huang Vanderbilt University Medical Center
REVIEW RETURNED	17-Feb-2023

GENERAL COMMENTS	All comments from previous review are addressed.
--